# *N*-Acetylcysteine Inhibits Patulin-Induced Apoptosis by Affecting ROS-Mediated Oxidative Damage Pathway

**DOI:** 10.3390/toxins13090595

**Published:** 2021-08-26

**Authors:** Jiayu Liu, Qi Liu, Jiahui Han, Jiayu Feng, Tianmin Guo, Zhiman Li, Fenyi Min, Ruyi Jin, Xiaoli Peng

**Affiliations:** College of Food Science and Engineering, Northwest A & F University, Yangling 712100, China; liujiayu1002@nwafu.edu.cn (J.L.); liuqi331@126.com (Q.L.); hanjiahui2021888@163.com (J.H.); fengjiayu@nwafu.edu.cn (J.F.); guotianmin666@nwafu.edu.cn (T.G.); lizhiman2001@nwafu.edu.cn (Z.L.); minfenyi@nwafu.edu.cn (F.M.); jinruyi@nwafu.edu.cn (R.J.)

**Keywords:** PAT, apoptosis, ROS, oxidative stress, mitochondrial oxidative phosphorylation

## Abstract

Patulin (PAT) belongs to the family of food-borne mycotoxins. Our previous studies revealed that PAT caused cytotoxicity in human embryonic kidney cells (HEK293). In the present research, we systematically explored the detailed mechanism of ROS production and ROS clearance in PAT-induced HEK293 cell apoptosis. Results showed that PAT treatment (2.5, 5, 7.5, 10 μM) for 10 h could regulate the expression of genes and proteins involved in the mitochondrial respiratory chain complex, resulting in dysfunction of mitochondrial oxidative phosphorylation and induction of ROS overproduction. We further investigated the role of *N*-acetylcysteine (NAC), an ROS scavenger, in promoting the survival of PAT-treated HEK293 cells. NAC improves PAT-induced apoptosis of HEK293 cells by clearing excess ROS, modulating the expression of mitochondrial respiratory chain complex genes and proteins, and maintaining normal mitochondrial function. In addition, NAC protects the activity of antioxidant enzymes, maintains normal GSH content, and relieves oxidative damage. Additionally, 4 mM NAC alleviated 7.5 μM PAT-mediated apoptosis through the caspase pathway in HEK293 cells. In summary, our study demonstrated that ROS is significant in PAT-mediated cytotoxicity, which provides valuable insight into the management of PAT-associated health issues.

## 1. Introduction

Patulin (PAT), a common mycotoxin, is a food contaminant produced by several fungal species such as *Byssochlamys*, *Aspergillus*, and *Penicillium* [1,2,3]. PAT is usually present in moldy food materials including apples and its derived products, vegetables, cereals, and cheese [4,5,6]. PAT is a polyketide lactone which reacts strongly with thiol groups [7,8]. Increasing toxicological evidence has suggested that human and livestock exposure to PAT produces several toxic effects, such as nephrotoxicity, hepatotoxicity, neurotoxicity, and gastrointestinal and dermal toxicities [3,9,10]. Several compounds were found to have protective effects on nephrotoxicity caused by PAT [3,11]. Our previous studies revealed that PAT exerts significant cytotoxicity via the production of intracellular ROS in HEK293 cells [12]. Moreover, studies have shown that ROS-mediated oxidative stress plays a major role in PAT-induced cytotoxicity. However, the underlying molecular mechanism of ROS generation in response to PAT exposure is still unclear.

Mitochondria are thought to be the cell’s “powerhouse” and play pivotal roles in many metabolic processes, including energy metabolism, oxidative stress, and apoptosis regulation [13,14,15]. The mitochondrial oxidative phosphorylation (OXPHOS) system involves the complexes of the electron transfer chain (ETC) and ATP synthase complex. In these compounds, complex I is NADH-ubiquinone reductase, complex II is succinate dehydrogenase, complex III is ubiquinol cytochrome c reductase, complex IV is cytochrome c oxidase, and complex V is ATP-synthase [1,16,17]. Cells performing OXPHOS generate the mitochondrial membrane potential, drive 90% of cellular ATP generation [18], and contribute significantly to cellular ROS. Therefore, regulation of OXPHOS is critical to maintain cellular homeostasis.

The research showed that that ROS play a key role in different cellular processes such as gene expression, energy metabolism, and cell signaling [19,20,21]. In normal physiological conditions, the antioxidant enzyme system usually regulates the production of ROS in cells to maintain a relatively balanced concentration [22]. Nevertheless, the dysregulation of the activity of the antioxidant enzyme system may cause an imbalance between production and scavenging of ROS, leading to higher concentrations of ROS, resulting in damage of various cellular components, such as lipids, enzymes, proteins, cell membranes, and DNA [23,24,25]. Furthermore, ROS overproduction ultimately leads to oxidative stress and triggering of the apoptotic pathway [26,27]. The balance of production and scavenging of ROS is considered key to maintaining healthy biological systems. Accordingly, antioxidant defense mechanisms that consist of enzymatic antioxidants are used to neutralize the extra ROS, including CAT (catalase), SOD (superoxide dismutase), GR (glutathione reductase), and GPx (glutathione peroxidases) [27,28,29]. Furthermore, exogenous antioxidants are often used to inhibit the rate of ROS production, such as NAC (*N*-acetylcysteine), vitamin C, vitamin E, polyphenols, and carotenoids [30].

NAC is an exogenous reactive oxygen scavenger and a precursor of reduced glutathione (GSH), which can directly react with oxidative substances to reduce ROS content in the body. Studies have shown that NAC destroyed thiolated proteins to release free mercaptans, which have better antioxidant activity than NAC, and promoted the synthesis of glutathione and reduced proteins [31]. It has been reported that NAC treatment effectively reduced the increase in ROS level of GES-1 cells caused by *Helicobacter pylori*, as well as inhibited the ROS-mediated activation of the PI3K/Akt pathway and DNA damage [32]. Through starvation and paraquat stress tests, NAC treatment was observed to upregulate the transcription levels of key enzymes that resist ROS attack, such as phospholipid peroxide glutathione peroxidase and catalase [33]. Additionally, NAC clearance of PM_2.5_-induced ROS can block cell apoptosis and restore the activity of the Nrf2 signaling pathway in human embryonic stem cells [34].

In the current research, we systematically elucidated the antioxidant effect of NAC in the regulation of cell survival toward PAT-induced intracellular ROS generation in HEK293 cells. More importantly, the potential molecular mechanisms, the relationship between the production and scavenging of ROS, and the effect of PAT on the apoptosis of HEK293 cells were investigated. Our research contributes to an understanding of the pathological role of ROS and provides theoretical support for NAC’s inhibition of PAT-induced cytotoxicity.

## 2. Result

### 2.1. PAT Induced Cytotoxicity and Apoptosis in HEK293 Cells, Diminished by NAC

Reports have revealed that ROS overproduction plays a key part in PAT-induced cytotoxicity [12]. Next, we tested the effects of NAC, an ROS scavenger, on cytotoxicity in response to PAT. As exhibited in Figure 1A, compared with PAT treatment group, the cell viability significantly increased by 102.07%, 389.48%, and 412.21% when pretreated with 2, 4, and 10 mM NAC. Therefore, 4 mM NAC was chosen for subsequent experiments as an optimum dose. We next examined the effect of NAC on cytotoxicity using the LDH assay. The result demonstrated that PAT treatment exhibited a remarkable increase in LDH leakage. In contrast, cotreatment with NAC significantly reduced the LDH leakage (Figure 1B).

Next, Hoechst 33342 (Beyotime Institute of Biotechnology, Beijing, China) staining was performed to measure cell apoptosis. When apoptosis occurs, the nucleus undergoes pyknosis. After staining with Hoechst 33342, the normal cells will appear normal blue under a fluorescence microscope, while the apoptotic cells will appear dense and heavily stained bright blue. As shown in Figure 1C, HEK293 cells exhibited a normal nucleus structure in the control and NAC alone treatment group. On the other hand, PAT treatment caused nuclear degradation and chromatin condensation in cells. Therefore, the Hoechst staining suggests apoptosis in PAT-treated cells due to nuclear morphological change. On the contrary, cotreatment with NAC obviously ameliorated the cytotoxicity induced by PAT.

To support this hypothesis, the apoptosis rate was further detected by annexin V and PI staining. Annexin V has a high affinity binding for PS, which can be utilized as a sensitive probe to explore PS exposed on the surface of cell membrane. The transfer of PS to the outer membrane may also occur during cell necrosis, but the cell membrane of necrotic cells is damaged, whereas the DNA of the cells can be stained by PI, while PI does not stain the early apoptotic cells [35]. On the scatter plot of bivariate flow cytometry, the upper right quadrant denotes annexin V+/PI+, which shows late-phase apoptotic cells, and the lower right quadrant refers to Annexin V+/PI−, which shows early-phase cells. These two quadrants were used to quantify the results in Figure 2B. As exhibited in Figure 2A,B, compared with the control group, PAT treatment significantly caused apoptosis. In contrast, cotreatment with NAC decreased the percentage of apoptosis cells. The results suggest that, in HEK293 cells, NAC effectively decreased the apoptosis caused by PAT.

### 2.2. NAC Reduced ROS Generation in Response to PAT Exposure

To clarify the role of NAC in oxidative stress induced by PAT, the changes in intercellular ROS levels were measured by DCFH-DA probe. As exhibited in Figure 3A,B, the mitochondria-derived ROS was assessed by a MitoSOX Red Mitochondrial Superoxide Indicator. Results indicated that, compared to the control group, PAT significantly increased the intensity of red fluorescence. Furthermore, PAT treatment resulted in a remarkable increase in intracellular ROS compared with the control treatment. Nevertheless, the ROS level was dramatically decreased in the presence of NAC (Figure 3C). The changes were significantly attenuated by NAC. All these results showed that NAC obviously reduced the generation of total and mitochondria-derived ROS in response to PAT exposure.

### 2.3. PAT Caused the Impairment of the Mitochondrial Respiratory Chain Complex Signaling Pathway

It is known that the mitochondrial respiratory chain (MRC) is a crucial origin of ROS in eukaryotic cells [36]. To elucidate whether the ROS production induced by PAT is caused by the disorder of MRC, the mRNA and protein expressions of the MRC complexes were determined. As shown in Figure 4, treatment with PAT induced a concentration-dependent reduction in complex III (UQCRQ) and complex V (ATP6, ATP8) mRNA levels of HEK293 cells. However, PAT increased the mRNA expression of complex IV (COX17) in a concentration-dependent manner. In addition, the 2.5 µM PAT treatment increased the mRNA transcription of complex I (NDUFA4) and complex II (SDHA), while the 7.5 µM PAT treatment decreased the mRNA transcription of complex II (SDHA). Similar results were observed in protein expressions of the MRC complexes (Figure 5A,B). These results suggested that PAT impaired the process of mitochondrial oxidative phosphorylation, leading to mitochondrial dysfunction and the generation of ROS.

### 2.4. NAC Improved the Disorders of the Mitochondrial Respiratory Chain Complex Induced by PAT

Mitochondria are relevant to the increase in ROS, mainly through the mitochondrial electron transport chain. To verify the role of NAC in the mitochondrial dysfunction caused by PAT, the mRNA and protein expressions of the MRC complexes were measured. As shown in Figure 6, PAT treatment obviously increased the expression of mRNA in complex II (SDHA) and complex IV (COX17). In addition, NAC improved the complex II (SDHA) and complex IV (COX17) gene expression. In contrast, PAT decreased complex III (UQCRQ) and complex V (ATP6, and ATP8) gene expression, whereas it increased after cotreatment with NAC. Western blotting showed similar results to mRNA expression (Figure 7A,B). Altogether, these results showed that NAC attenuated the mitochondrial dysfunction caused by PAT via the MRC complex pathway.

### 2.5. NAC Protected against PAT-Induced GSH Depletion and Improved Antioxidant Enzyme Activities

Accumulating evidence has indicated that glutathione metabolism and other antioxidant enzymes are the most critical cellular defense mechanisms for protecting against oxidative stress damage [37]. To explore the positive effect of NAC, the levels of GSH and GSSG were examined. PAT treatment decreased GSH level and increased GSSG level significantly compared with the control group. In contrast, NAC pretreatment exhibited an opposite effect compared to the PAT group (Figure 8A,B). Consistently, the ratio of GSH to GSSG was significantly increased in the PAT plus NAC treatment group (Figure 8C). The activities of CAT, SOD, GPx, and GR were reduced notably after PAT treatment, whereas cotreatment with NAC dramatically increased their activities (Figure 8D–G). Taken together, these findings suggested that NAC can reduce the GSH depletion and restore the decline in antioxidant enzyme activities induced by PAT.

### 2.6. NAC Inhibited PAT-Induced Mitochondrial Dysfunction and Caspase-Dependent Apoptotic Pathway

The results showed that MMP was decreased in the cells treated by PAT; the green fluorescence was enhanced while the red fluorescence was weakened. However, the value of the MMP was attenuated by cotreatment with NAC (Figure 9A,B). We next examined the ATP level using an ATP assay kit. As exhibited in Figure 9C, PAT prominently reduced ATP production when compared with control cells. In contrast, it exhibited a notable increase in ATP levels upon pretreatment with NAC. Taken together, these results revealed that NAC suppressed PAT-induced mitochondrial dysfunction.

Caspases represent a family of proteolytic enzymes that function as initiators and executors of apoptosis [38]. Our previous research showed that apoptosis induced by PAT was related to caspase cascade activation [39]. Therefore, we assessed the activities and mRNA expressions of caspase 3, caspase 8, and caspase 9. As exhibited in Figure 9D, the activities of caspases 3, 8, and 9 obviously increased after PAT treatment, whereas pretreatment with NAC caused a significant reduction in caspase 3, 8, and 9 activities. Consistent with the enzymatic activities, the mRNA levels of caspase 3, 8, and 9 were greatly upregulated after PAT treatment, whereas they were significantly attenuated by NAC (Figure 9E). Collectively, these results clearly indicate that NAC suppressed PAT-induced apoptosis through the caspase-dependent pathway.

## 3. Discussion

Studies have evidenced that PAT exerts cytotoxicity through accelerating the production of intracellular ROS, leading to apoptosis [11,40]. Mitochondrial oxidative phosphorylation is the main source of ROS, which is made up of a sequence of respiratory complexes (complexes I–V) [41,42], and alterations in oxidative phosphorylation (OXPHOS) result in the generation of ROS, further inducing oxidative stress and mitochondrial dysfunction [13,43,44,45].

In the present research, we observed a significant change in the activities of mitochondrial respiratory complexes in response to PAT. In detail, the levels of complex III (UQCRQ) and complex V (ATP6, ATP8) obviously decreased, whereas that of complex IV (COX17) increased after PAT treatment; however, the levels of complex I (NDUFA4) and complex II (SDHA) first increased and then decreased with an increase in the concentration of PAT. Moreover, total and mitochondrial ROS increased greatly.

Our previous studies have indicated that PAT can induce apoptosis [12,39]. The current study focused on the protective role of NAC, a water-soluble antioxidant agent, on PAT-induced apoptosis in HEK293 cells. Studies have shown that NAC can scavenge free radicals, increase the level of cellular glutathione, and decrease depolarization of the mitochondrial membrane [46,47]. In this study, the production of total and mitochondrial ROS was notably attenuated by NAC treatment. Briefly, NAC treatment elevated the ATP contents and MMP values compared with the PAT treatment group. Moreover, our studies found that inhibition of antioxidant NAC improved mitochondrial oxidative phosphorylation through the regulation of MRC complexes.

It is well known that caspases are involved in apoptosis regulation. In the current research, we discovered a significant increase in the enzyme activities of caspases 3, 8, and 9 in response to PAT exposure, whereas NAC effectively inhibited their activities. Consistent with this result, NAC led to a significant decrease in the mRNA levels of caspases 3, 8, and 9. These results indicated that NAC reduce PAT-induced apoptosis by inhibiting the activation of caspases 3, 8, and 9.

To protect cells from oxidative damage, cellular defense mechanisms will produce numerous antioxidant enzymes (SOD, CAT, GR, GPx) and antioxidants such as GSH to neutralize superoxide radicals. It is well known that GSH can be oxidized to GSSG. Consistent with our previous studies, we discovered that intracellular ROS overproduction induced by PAT is accompanied by GSH depletion and increased GSSG content [39]. However, in the current research, we noticed marked increases in the ratio of GSH to GSSG involved in groups administered NAC and PAT. To further verify the protective effects of NAC, we measured the activities of SOD, CAT, GR, and GPx. Research has shown that the cellular antioxidant enzyme SOD plays a central role in scavenging the superoxide ion by speeding up its dismantlement [37], whereas CAT is a familiar enzyme which can convert one hydrogen peroxide molecule into two independent water molecules through catalyzing peroxide reactions [48]. GR plays a crucial part in the neutralization of hydroperoxides; moreover, GPx can scavenge hydrogen peroxide and other peroxides using GSH as the reducing agent [49,50,51]. The results obtained in the present research also proved that pretreatment with NAC obviously reversed the decrease in activities of SOD, CAT, GR, and GPx induced by PAT. Taken together, our results clearly suggest that NAC can obviously attenuate the PAT-induced oxidative damage through decreasing the activity of antioxidant enzymes.

At present, most studies on the toxicity of PAT have been conducted in vitro. Existing in vivo experiments have shown that feeding 10 mg/kg PAT for 4 days in BALB/c mice significantly increased the levels of serum urea and LDH, and some pathological changes including renal tubule swelling, vacuolar degeneration, and protein casting occurred in the kidneys [3]. BALB/C mice were fed 2 mg/kg PAT for 7 consecutive days and then injected intraperitoneal with the same concentration for 3 days; the production of ROS in liver tissue measured by DCFH-DA increased by 288%, the expression of oxidative markers such as SOD and CAT was downregulated, and the expression of apoptotic enzymes p53 and caspase 3 was increased. The results indicated that PAT promoted oxidative stress and apoptosis in the liver [52]. The toxicity of PAT demonstrated by the results of this study is consistent with the above in vivo experiments, which lays a foundation for further study of the effect of NAC on the ROS-mediated oxidative damage pathway induced by PAT in vivo.

## 4. Conclusions

In summary, our study demonstrated that PAT can promote mitochondrial dysfunction and oxidative stress in HEK293 cells through increasing intracellular ROS production and regulating mitochondrial function. Furthermore, treatment with the antioxidant NAC significantly suppressed oxidative damage and apoptosis in response to PAT exposure. Together, these findings can help to understand fundamental molecular mechanisms underlying ROS production and apoptosis for the treatment of PAT, as well as provide a novel treatment strategy for PAT-induced cytotoxicity.

## 5. Materials and Methods

### 5.1. Chemicals and Reagents

*N*-Acetyl-1-cysteine (NAC, purity ≥ 98%) and patulin (PAT, purity ≥ 99%) were supplied by Sigma-Aldrich (St. Louis, MO, USA). The LDH-assay kit, ROS assay kit, total SOD assay kit, annexin V–FITC apoptosis assay kit, Hoechst 33342 dyes, GSH and GSSG assay kit, GR assay kit, total GPx assay kit, ATP assay kit, mitochondrial membrane potential (MMP) assay kit, the assay kits of caspase 3, 8, and 9, and BCA protein-assay kit were obtained from Beyotime Institute of Biotechnology (Beijing, China). MitoSOX Red Mitochondrial Superoxide Indicator was purchased from Invitrogen Corporation (St. Louis, MO, USA). The CAT test kit was obtained from Jiancheng Bioengineering Institute (Nanjing, Jiangsu, China). The Ultrapure RNA kit, Super RT cDNA kit, and UltraSYBR mixture were purchased from CWBIO (Beijing, China).

### 5.2. Cell Cultivation and Treatment

HEK293 cells were purchased from Zhongqiao Xinzhou Biotechnology Company (Shanghai, China), maintained in Dulbecco’s modified Eagle medium (DMEM) supplemented with 10% FBS and 1% penicillin–streptomycin. At the temperature of 37 °C, the cells were cultivated in a humidified incubator with 5% CO_2_ and 95% air. When the HEK293 cells were grown to approximately 80% confluence, the cells were treated with 7.5 μM PAT for 10 h, or the cells were pretreated with NAC at different concentrations (0, 2, 4, and 10 mM) for 3 h.

### 5.3. Cell Viability Assays

MTT assay and LDH leakage were used to determine the cell viability according to the manufacturer’s instructions. In short, 1 × 10^5^ HEK293 cells were plated in a 96-well culture plate. Then, 24 h later, 7.5 μM PAT was added to the cells, or they were pretreated with different concentrations of NAC (0, 2, 4, and 10 mM) for 3 h, before treating with PAT for 10 h. Thereafter, 10 μL of MTT reagent (5 mg/mL in sterile PBS) was added to every well and incubated at 37 °C for 4 h. Next, the medium was discarded, and the sample was supplemented 150 μL of DMSO to dissolve formazan crystals. A microplate reader (Bio-Rad 680) was used to measure the absorbance at a wavelength of 570 nm. All experiments were repeated five times.

The LDH assay kit was used to assess LDH leakage according to the manufacturer’s instructions. PAT (7.5 μM) was added to cells, or 4 mM NAC was added to pretreat the cells for 3 h, before being treated by PAT for 10 h. The absorbance was measured using a microplate reader (Bio-Rad 680) at 490 and 630 nm wavelengths.

### 5.4. Hoechst 33342 Staining

Hoechst 33342 staining was used to examine cell apoptosis. After treating with 7.5 μM PAT or pretreating with 4 mM NAC for 3 h, the cells were treated with PAT for 10 h. The cells were washed twice with PBS (pH 7.4) and then incubated with Hoechst 33342 dye (10 μg/mL) in the dark for 20 min (at the condition of 37 °C). The cells were observed under an inverted fluorescence microscope (Lecia DMI8, Germany).

### 5.5. Apoptosis Assessment

Flow cytometry with an annexin V–FITC/PI apoptosis detection kit was used to detect the cell apoptosis ratio. Briefly, the HEK293 cells were seeded in a six-well plate, treated as described above. Then, the cells were trypsinized, collected, and washed twice with ice-cold PBS, before centrifuging at 1000× *g* for 5 min. The collected cells were resuspended with 195 μL of annexin V–FITC binding buffer, and stained with 5 μL of annexin V–FITC for 10 min and 10 μL of propidium iodide (PI) for 5 min in the dark. Afterward, the cells were immediately analyzed by flow cytometry (FACS Calibur, Franklin Lakes, NJ, USA).

### 5.6. Measurement of Mitochondrial Membrane Potential

The JC-1 fluorescent probe was used to assess the mitochondrial membrane potential. In short, after treatment as described above, the cells were cultured with the JC-1 dye for 20 min at a temperature of 37 °C. After that, the stained cells were washed twice with JC-1 dyeing buffer (1×). Then, the samples were observed under inverted fluorescence microscopy. The red/green fluorescence values were determined using a multifunctional microplate analyzer (Tecan, Infinite M200 Pro, Männedorf, Switzerland).

### 5.7. Measurement of ATP Level

An ATP assay kit was used to determine the adenosine triphosphate content according to the manufacturer’s protocols. After treatment as above, the cells were broken down and centrifuged at 12,000× *g* for 5 min at a temperature of 4 °C to collect supernatant. Then, 10 µL of supernatant was mixed with 100 µL of ATP detection solution in black 96-well plates. After incubation for 5 min, the change in ATP levels was detected by chemiluminescence using a multifunctional microplate reader (Tecan, Infinite M200 Pro, Männedorf, Switzerland).

### 5.8. Measurement of Total ROS and Mitochondrial ROS

Total ROS in HEK293 cells was evaluated using a 2,7-dichloroflfluorescein diacetate (DCFH-DA) probe. After treatment as above, the cells were cultured using a 10 μM DCFH-DA probe for 20 min at 37 °C. The fluorescence was observed under a fluorescent microscope. The fluorescence intensity was measured using a multifunctional microplate analyzer at 488 nm excitation wavelength and 520 nm emission wavelength.

To measure the mitochondrial ROS, the MitoSOX Red Mitochondrial Superoxide Indicator was used according to the manufacturer’s instructions. After treatment as described above, the working probe fluid was added to react with cells for 10 min at a temperature of 37 °C. Then, a laser confocal microscope (Nikon, A1) was used to obtain the fluorescence images.

### 5.9. Analysis of GSH and GSSG Contents

GSH and GSSG assay kits were used to measure GSH and GSSG levels. Briefly, after treatment, the cells were washed with ice-cold PBS, before being centrifuged at 600× *g* for 5 min. Next, the cell volume was evaluated. Then, a protein-removal regent at 10 times the cell volume was added, and the cells underwent two freeze–thaw cycles in liquid nitrogen and a 37 °C water bath. The cells were centrifuged at 10,000× *g* for 10 min at 4 °C to collect supernatant. Afterward, the GSH and GSSG content was measured according to the manufacturer’s instructions. The absorbance at 412 nm was assayed using a multifunctional microplate reader.

### 5.10. Activities of SOD, CAT, GR, and GPx

Briefly, cells were treated as above, and the supernatants were collected after lysing and centrifuging at 12,000× *g* at 4 °C for 10 min. The protein concentration was detected using a BCA protein assay kit.

Activity of SOD: A total superoxide dismutase assay kit with WST-8 was used to measure SOD activity. The cell supernatants were incubated with WST working solution and enzyme working solution at room temperature for 20 min. A multifunctional microplate reader was used to measure absorbance at 450 nm.

Activity of CAT: A catalase assay kit was used to detect the CAT content according to our previous studies [53].

Activity of GR: GR activity was measured using a glutathione reductase assay kit with NADPH as the reducing agent. The amount of yellow-TNB was used to reflect GR activity, produced by the reaction between GSH and DTNB. A multifunctional microplate reader was used to measure absorbance at 412 nm.

Activity of GPx: GPx activity was determined using a total glutathione peroxidase assay kit. GPx activity was measured on the basis of the decrease in NADPH in the presence of GPx. A multifunctional microplate reader was used to measure absorbance at 340 nm.

### 5.11. Caspase Activity Assay

The corresponding caspase activity assay kits were used to assay the activities of caspases 3, 8, and 9 according to the manufacturer’s instructions. In short, the cells were treated as described above. The cells were collected and lysed with lysis buffer, before being centrifuged at 16,000× *g* for 15 min at a temperature of 4 °C. Afterward, the supernatants were incubated with 5 μL of Ac-DEVD-pNA, Ac-IETD-pNA, or Ac-LEHD-pNA in the dark at 37 °C for 2 h. The absorbance at 405 nm was determined using a microplate reader (Bio-Rad 680).

### 5.12. Quantitative Real-Time Reverse Transcription PCR (qRT-PCR)

After treatment as mentioned above, TRIzon Reagent was used to extract total mRNA from HEK293 cells. The RNA was reverse-transcribed into cDNA using the Super RT cDNA kit according to the manufacturer’s protocol. In the qRT-PCR analysis, cDNA amplification was performed using a two-step UltraSYBR mixture and was detected by the IQ5 Multicolor Real-Time PCR Detection System (Bio-Rad). The primer sequences are provided in Appendix A.

### 5.13. Western Blot Analysis

The Western blot assay was processed as described previously (Wang et al., 2018). Briefly, the cells were broken down using a lysis buffer supplemented with 1 mM PMSF and then homogenized for 10 min (11,000× *g*, 4 °C). The supernatants were collected and then denatured with 4× loading buffer at 95 °C for 10 min. The BCA protein assay kit was used to determine the protein concentration. Subsequently, the proteins were separated by 10–15% SDS-PAGE, and then transferred onto a PVDF membrane, which was blocked with 5% nonfat milk dissolved in TBST at room temperature for 2 h. Next, they were incubated with primary antibodies (1:1000), including anti-NDUFA4, anti-SDHA, anti-UQCRQ, anti-COX17, anti-ATP6V1B2, anti-ATP8B2, and anti-β-actin. After incubation overnight at 4 ℃, the blots were washed three times for 20 min before incubating with the secondary antibody at room temperature for 2 h. The blots were detected using a hypersensitive ECL chemiluminescence reagent (Beyotime) and exposed to a Chemiluminescent imaging system (Bio-Rad). Image J software was used to quantitate the protein intensity.

### 5.14. Statistical Analysis

All data are presented as means ± standard deviations (SD) from at least three independent experiments. Significant differences were analyzed with one-way factorial analysis of variance (ANOVA) and Tukey’s test (SPSS 19.0). A value of *p* < 0.05 represented a statistically significant difference.

## Figures and Tables

**Figure 1 toxins-13-00595-f001:**
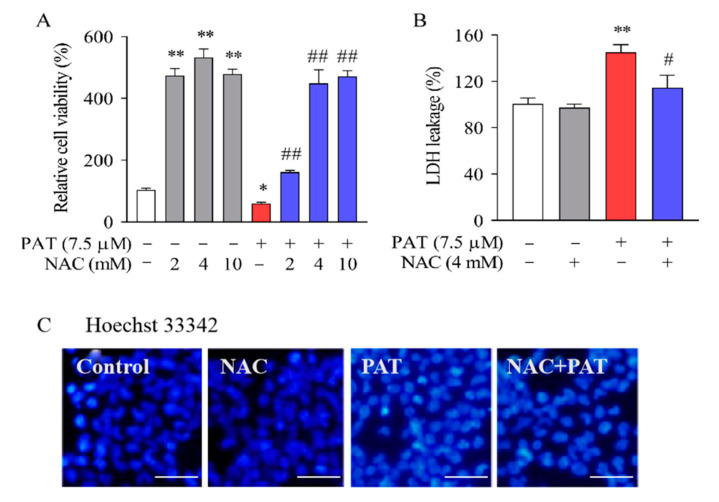
In HEK293 cells, NAC diminished the cytotoxicity and apoptosis caused by PAT. HEK293 cells were treated by 7.5 μM PAT with or without 4 mM NAC. (**A**) The cell viability of different groups was measured using an MTT assay (means ± SD, *n* = 5). (**B**) The cytotoxicity of different groups was measured using the LDH leakage assay (means ± SD, *n* = 3). (**C)** Hoechst 33342 was used to stain HEK293 cells. The images are representative of three independent experiments. ^#^ *p* < 0.05 and ^##^ *p* < 0.01 compared with the PAT treatment group; * *p* < 0.05 and ** *p* < 0.01 compared with the control treatment group.

**Figure 2 toxins-13-00595-f002:**
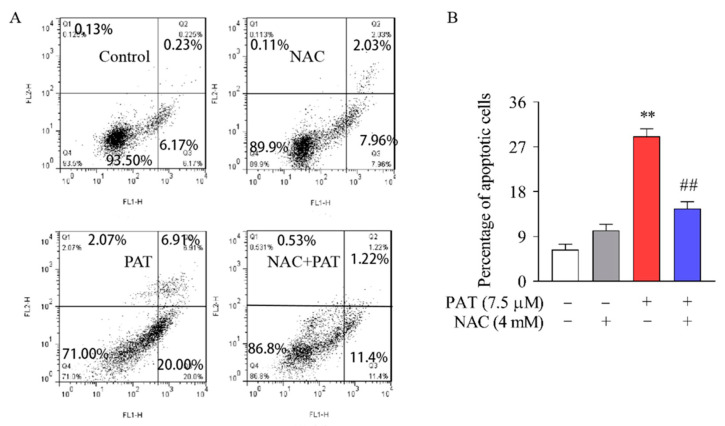
In HEK293 cells, NAC diminished cytotoxicity and apoptosis caused by PAT. HEK293 cells were treated by 7.5 μM PAT with or without 4 mM NAC. (**A**) Apoptosis was analyzed by flow cytometry with annexin V–FITC and PI dual staining. FL1 refers to annexin V–FITC and FL2 refers to PI. The lower right quadrant refers to annexin V–FITC-stained cells and the upper right quadrant refers to PI and annexin V–FITC-dual-stained cells. (**B**) Quantification of apoptosis rates (means ± SD, *n* = 3). ^##^ *p* < 0.01 compared with the PAT treatment group; ** *p* < 0.01 compared with the control treatment group.

**Figure 3 toxins-13-00595-f003:**
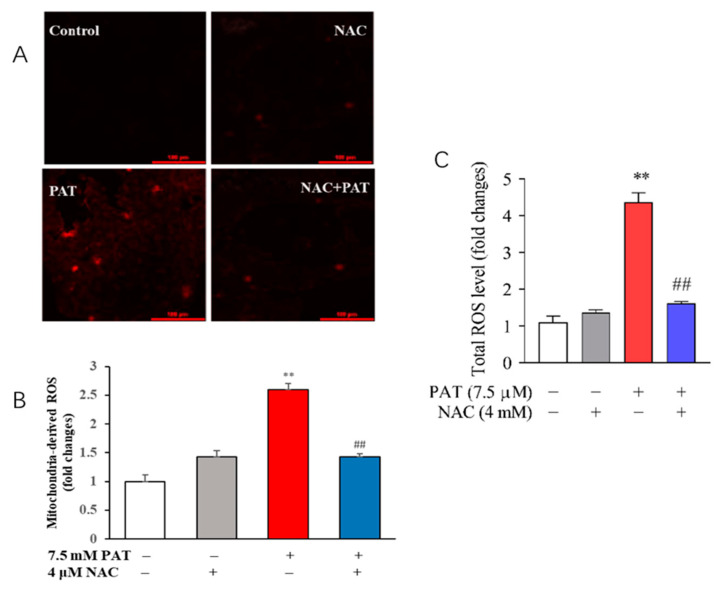
NAC reduced ROS reproduction in response to PAT exposure. HEK293 cells were treated by 7.5 μM PAT with or without 4 mM NAC. (**A**) Fluorescent image. The mitochondrial ROS level was measured with MitoSOX Red Mitochondrial Superoxide Indicator. The images are representative of three independent experiments. (**B**) Fold changes of mitochondrial ROS (means ± SD, *n* = 3). (**C**) The DCFH-DA probe was used to assess total ROS levels (means ± SD, *n* = 3). ^##^ *p* < 0.01 compared with the PAT treatment group; ** *p* < 0.01 compared with the control treatment group.

**Figure 4 toxins-13-00595-f004:**
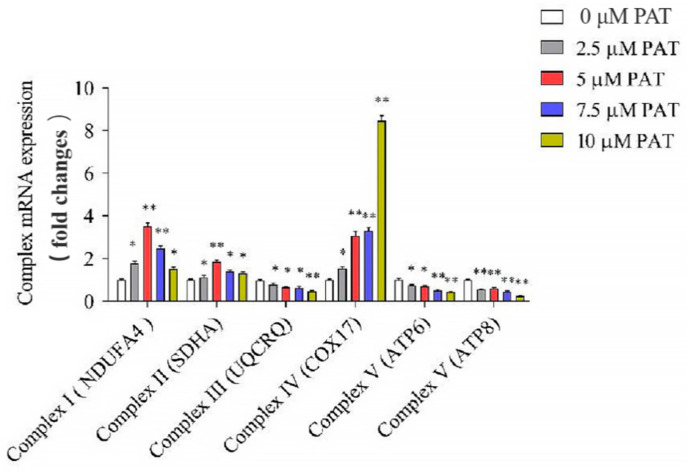
The gene levels of the mitochondrial respiratory chain complex. PAT caused the impairment of the mitochondrial respiratory chain complex signaling pathway. After 10 h treatment of HEK293 cells with different concentrations of PAT, the mRNA expression was detected by real-time PCR (means ± SD, *n* = 3). * *p* < 0.05 and ** *p* < 0.01 compared with the 0 μM PAT group.

**Figure 5 toxins-13-00595-f005:**
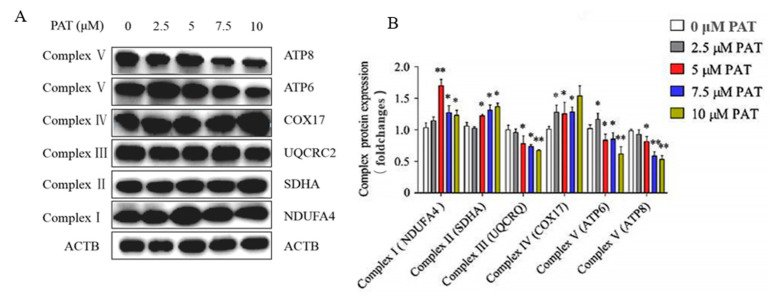
PAT caused the impairment of the mitochondrial respiratory chain complex signaling pathway. Different concentrations of PAT were used to treat HEK293 cells for 10 h. (**A**) Western blot detection of respiratory chain complex protein expression after PAT treatment at different concentrations. β-Actin served as an internal reference. The images are representative of three independent experiments. (**B**) Mitochondrial respiratory chain complex protein expressions (means ± SD, *n* = 3). * *p* < 0.05 and ** *p* < 0.01 compared with the 0 μM PAT group.

**Figure 6 toxins-13-00595-f006:**
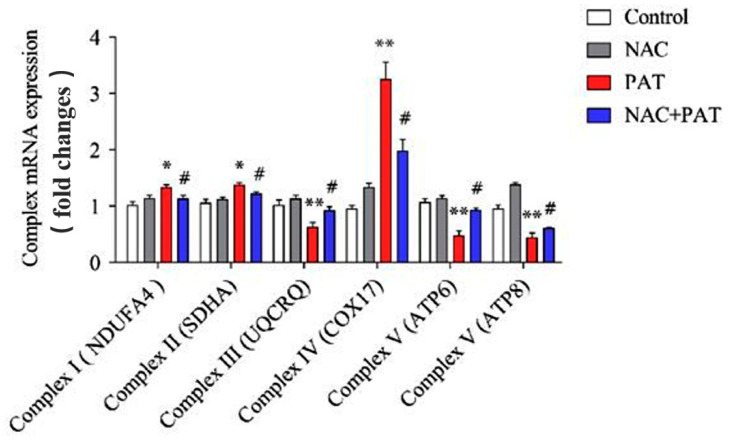
The gene levels of the mitochondrial respiratory chain complex. NAC improved the disorders of the mitochondrial respiratory chain complex induced by PAT. HEK293 cells were treated by 7.5 μM PAT with or without 4 mM NAC. The mRNA expression was determined by real-time PCR (means ± SD, *n* = 3). ^#^ *p* < 0.05 compared with the PAT treatment group; * *p* < 0.05 and ** *p* < 0.01 compared with the control treatment group.

**Figure 7 toxins-13-00595-f007:**
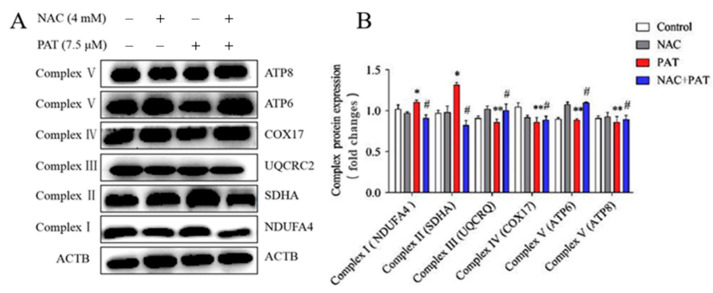
NAC improved the disorders of the mitochondrial respiratory chain complex induced by PAT. HEK293 cells were treated by 7.5 μM PAT with or without 4 mM NAC. (**A**) Western blot detection of respiratory chain complex protein expression after different treatments. β-Actin served as an internal reference. The pictures are representative of three independent experiments. (**B**) Mitochondrial respiratory chain complex protein expressions (means ± SD, *n* = 3). ^#^ *p* < 0.05 compared with the PAT treatment group; * *p* < 0.05 and ** *p* < 0.01 compared with the control treatment group.

**Figure 8 toxins-13-00595-f008:**
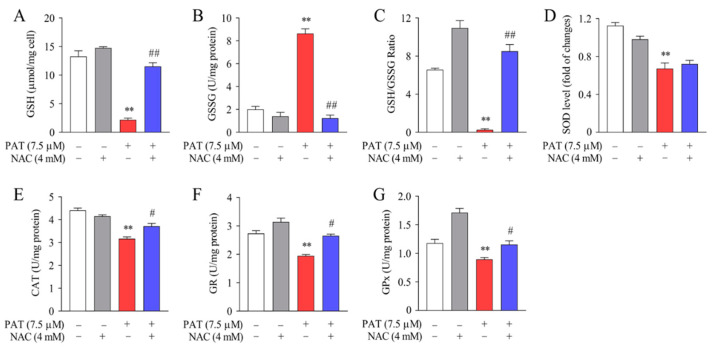
NAC protected against PAT-induced GSH depletion and improved antioxidant enzyme activities. HEK293 cells were treated by 7.5 μM PAT with or without 4 mM NAC. (**A**) Content of GSH; (**B**) content of GSSG; (**C**) ratio of GSH to GSSG; (**D**) activity of SOD; (**E**) activity of CAT; (**F**) activity of GR; (**G**) activity of GPx. All the above were measured using different assay kits (means ± SD, *n* = 3). ^#^ *p* < 0.05 and ^##^ *p* < 0.01 compared with the PAT treatment group; ** *p* < 0.01 compared with the control treatment group.

**Figure 9 toxins-13-00595-f009:**
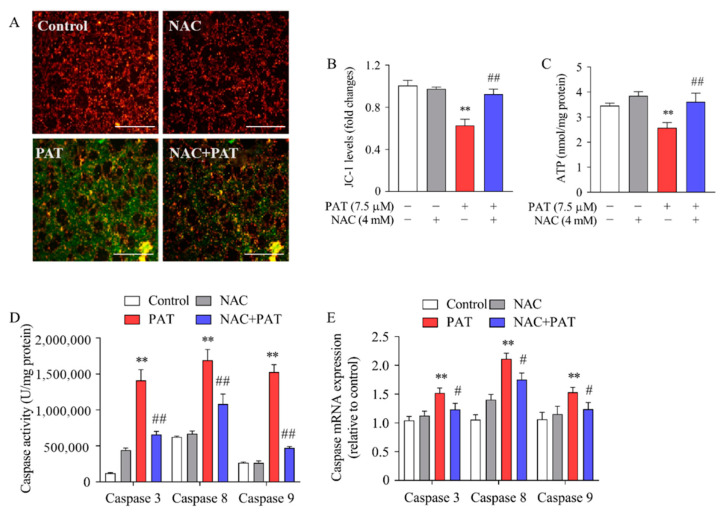
NAC inhibited PAT-Induced mitochondrial dysfunction and apoptosis dependent on caspase pathway. HEK293 cells were treated by 7.5 μM PAT with or without 4 mM NAC. (**A**,**B**) JC-1 staining was used to analyze MMP. Fluorescent images are representative of three independent experiments. JC-1 levels came from the ratio of red/green fluorescence (means ± SD, *n* = 3). (**C**) ATP levels of different treatments were detected using an ATP assay kit (means ± SD, *n* = 3). (**D**) The activities of the caspase 3, caspase 8, and caspase 9 were measured as described in Section 5 (means ± SD, *n* = 3). (**E**) Real-time PCR analyses for the expression of caspase 3, caspase 8, and caspase 9 (means ± SD, *n* = 3). ^#^ *p* < 0.05 and ^##^ *p* < 0.01 compared with the PAT treatment group; ** *p* < 0.01 compared with the control treatment group.

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
