# Peer review of "N*-Acetylcysteine Inhibits Patulin-Induced Apoptosis by Affecting ROS-Mediated Oxidative Damage Pathway"

_toxins, 2021, doi:10.3390/toxins13090595_

Round 1

Reviewer 1 Report

The manuscript explains describes the Involvement of ROS-mediated oxidative damage pathway in PAT-induced apoptosis. Manuscript needs certain revision before acceptance for publication:

  1. The author showed PAT induced mitochondrial stress via ROS generation that impacts on MMP. Furthermore, a preventive effect of NAC was demonstrated to counter PAT-induced oxidative stress and subsequent targets. Can the author quantify CytoC, pJNK, Bax, and Bcl2 levels as they are the most prominent markers of mitochondrial stress?
  2. For DCFDA staining, provide the fluorescent image which is missing in Figure 3.
  3. Provide the fold changes for the blots complex IV, III, II, and I (figure 4B). Also, provide the fold changes (densitometry analysis) for the blots of complex IV and complex (figure 5B).
  4. In the title of the manuscript exchange PAT with patulin.
  5. In figure 1 A, I am wondering how cell viability increase to 102.07, 389.48, and 412.21% after treatment with NAC (Line 74). Even the representation is not looking good. Address the issue.
  6. In abstract (line 10) remove 4 mM before NAC also remove 7.5 uM before PAT (line 11).
  7. Rewrite few sentences e.g “It has been …..in vivo models (line 32-33). In order to investigate ………..with control group (line 160-162). Previous studies ……… caspases cascade (186-187). Additionally, NAC ……………………….treatment group (221-222).
  8. Address the typo issue line no 74, 81, 121, and 237.
  9. Correct the reference as (43-45) (line 243).

Reviewer 2 Report

Good research about the important role of ROS played in PAT-mediated cytotoxicity.

Reviewer 3 Report

Dear Editor,

Thank you for sending the manuscript for reviewing. The current work unravels the underlying mechanism for PAT toxicity. PAT is toxic a fungal metabolite present in certain food and fruits.

. The authors conducted many experiments. I don’t see the innovative in the work as this is already published in the literature. Perhaps only the author used different type of cells. Further, exposure to PAT is usually chronic i.e. very low doses on long term. This makes the work to matching the real situation for human exposure.

Line 31, PAT is causing dermal toxic effect ?

Line 61, add the full name for NAC in the text

Line 68 ….. “provides a theoretical support against PAT toxin”, what do you mean ?

Reviewer 4 Report

toxins-1283325-peer-review-v1

This manuscript is interesting in that it seeks to elucidate the molecular mechanism(s) of PAT-induced cytoxicity using an in vitro HEK293 cell culture model.  The major flaws of this paper are: (1) requires extensive editing /proof-reading for both English usage and grammar- too many run-on long sentences and incorrect usage of terms, (2) insufficient explanations for methods and some results, (3) does not clearly state experimental controls/n= how many independent experiments performed/n= how many biological and/or technical replicates per experiment, and (4) Figures and Figure legends need clarity.

Comments and Concerns:

  • requires extensive editing /proof-reading for both English usage and grammar- too many run-on long sentences and incorrect usage of terms.  Too many examples in the text:
    1. Line 10-14 in abstract- … We further investigated that the survival.. Should be- we further investigated the role of N-acetylcysteine (NAC), a ROS scavenger, in promoting the survival of PAT-treated HEK293 cells. Then you could start a new sentence stating what the study found rather than one long run-on confusing sentence.
    2. Same sentence… relief oxidative damage… should be relieve oxidative damage.
  • Line 25- introduction- the mycotoxin patuilin (PAT) should be patulin.
  1. Line 28-29… with highly reactive toward thiol groups… should this be which is highly reactive towards thiol groups?
  2. Line 32--- it has been shown that several compounds could the protective effect- Not sure what you are saying here. If it has been studied, it should be past tense…. Several compounds were protective?... Please keep the past tense if it has been shown/done already and not use could.
  3. Line 74- pro-treated… you mean pre-treated..
  • Line 81.. Nonetheless, Nonetheless ? should only be on. Please be careful to have spacing correctly
  • 37°C should be 37 °
  1. Line 79 Hoechst 33,342 staining. Should be Hoechst 33342- no comma, please check through the text
  2. Line 277: 1x 105 HEK293, should be 1 x 105

  • insufficient explanations for methods and some results, does not clearly state experimental controls/n= how many independent experiments performed/n= how many biological and/or technical replicates per experiment.

  1. lines 85-89: flow cytometry to measure apoptosis rate. Refers to Fig. 2A and B and states that pre-treatment with NAC with PAT reduces apoptosis but doesn’t really explain the figure. Doesn’t explain what FL1 and FL2 are- annexin-FITC and Propidium iodide? Why are both markers important for apoptosis. Should state that dual staining with both markers indicate apoptosis in the results text instead of just in the figure legend. What quadrant are readers supposed to focus on? It is not written for the general audience to be easily decipherable. What quadrant is important- the top right double positive quadrant? How many cells were read- 10,000? Are the percentage positive cells from 10,000 or equivalent cells read from each of the cell treatments?

  1. Lines 79-83 and Fig 1C: regarding the Hoechst staining. In the results, the authors are using the images shown in Fig. 1C to state that there is apoptosis present in PAT treatment and NAC treatment represses the phenotype. I think Fig. 1C is not a strong argument for supporting their conclusion in the Results text. The authors show only 4 “representative” images. There is no quantification associated with these images. If they had quantified the Hoechst staining images, then they need to state how many cells per field, how many fields, how many total cells, and how many “apoptotic” nuclei per 100 cells (or something like that). I think if they leave Fig. 1C in, then they can say that the Hoechst staining suggests apoptosis in PAT treated cells due to nuclear morphological changes and further experiments measuring annexin V and PI staining were used to support this hypothesis.

  1. Methods section 5.3- cell viability assays- line 286-287, does this mean that you read each well 5x? how many biological or technical replicates per control/treatment were there per independent experiment? Reading one well 5x does not count for statistical and biological relevance.

  1. Hoechst staining- 5.4. It is Hoechst 33342 not 33,342. Was there any quantification of “apoptotic” nuclei for each condition, if so state how many cells per field? How many images were taken per condition?

  1. Section 5.5- HEK293 cells were digested- what is meant by this? You mean trypsinized? Please re-write this section to be transition clearly and not have long run on sentences. Please explain what staining refers to FL-1 vs FL-2. How many cells analyzed- 10,000?

  1. Section 5.6- it is stated that the ratio of red/green fluorescence was measured using the Tecan plate reader. In Figure 7B, the JC-1 levels were stated with fold changes. What fold changes- I am a bit uncertain of this meaning? Fold changes generally is used to denote in comparison to some factor? Shouldn’t this label be ratio of red/green fluorescence? Please clarify.

  1. Section 5.7- measurement of ATP level. How were the ATP levels detected? Luminescence, fluorescence, et?

  1. Section 5.8, if you are measuring fluorescent signal, then Figure 3B y-axis label should be Fluorescent signal/intensity? Why fold changes for total ROS levels? For MitoSox red experiment, in Figure 3A, the authors only show 4 “representative” images then states that there is less signal in the NAC+PAT.  The red signal intensity needs to be quantified from these images with means+SD or means+SEM. Please state how many independent, technical/biological replicates were used to determine the experimental result.

  1. Section 5.9, please check spelling and formatting. Add details of independent experiments and biological replicates.

  1. Section 5.10, same as above with n= independent and biological/technical replicates

  • insufficient explanation for figure legends/Figures

  1. Please add relevant information for all figure legends, n= independent experiments; n=biological/technical replicates used to determine means/SD. Add which statistical method was used to determine p values for all figures. Please explain a little bit of the results for the figures instead of saying Proteins, MMP-analyzed by JC-1 staining, etc.

  1. Figure 1C, I don’t think the Hoechst stain is necessary.

  1. Figure 2A, please either relabel FL-1 or FL-2 to be either Annexin V-FITC or PI, this will correspond better with the figure legend where these two terms were used instead of FL-1 or FL-2. Please clearly define the double positive quadrant in 2A that will be used to quantify for (B). Please add how many cells were screened so we can determine (b) what percentage of cells were double positive. 10,000 is usually the minimal number of cells that need to be screened via flow cytometry.

  1. Figure 4. This figure would make more sense If the authors move Fig.4A to where Fig 4B is currently and move Figure 4B to 4A is. Figure 4A- real time PCR: For complex III, V, V where it is claimed that there is a statistically significant concentration dependent decrease in gene expression with PAT treatment. What is the relative to control? GAPDH or Actin? Please put that on the Y-axis label instead of control. It is also hard to believe that there is a statistical difference for complex III, V, and V because on that graph, the changes look minimal in comparison to the changes to complex IV. Maybe you should split this graph into two- so the readers can see the differences more clearly. Fig 4B- The western blot at least shown here and developed via chemiluminescence is too over-exposed. Since the pixels are saturated already, one cannot tell the true difference using that exposure. Was the quantification for Fig. 4C extracted from this exposure? The authors should use a lower exposure/load less in order to do a proper densitometry analysis. Were the levels for each antibody normalized against the control actin signal?

  1. For Figure 5, same comments as for Figure 4.

Round 2

Reviewer 4 Report

The authors have revised the manuscript and it is a definite improvement over the previous version.  The manuscript reads and flows much better with more information in the figures. Minor edits are needed.

Comments:

  1. Abstract: line 13-14 is not a complete sentence, suggest Additionally, NAC protects the activity of antioxidant...
  2.  Line 33- there should be no peirod after PAT, the period should be only after [3,10].
  3. Line 80- do not capitalize Staining
  4. Line 89, should be ... further experiments with Annexin V..
  5. All figures... should be .. representative of .... indepedent experiments not experiment.
  6. Line 117, suggest instead fo to clear the role.. To clarify the role of NAC...
  7. Figures 4 and 6, These figures only show the mRNA levels. Please limit the description to say that the mRNA levels were measured by real-time PCR. Do not add protein levels/western blot in the description.
